# How Do Zooplankton Communities Respond to Environmental Factors across the Subsidence Wetlands Created by Underground Coal Mining in the North China Plain?

Yue Liang [1], Jianjun Huo [2], Weiqiang Li [1], Yutao Wang [1], Guangyao Wang [1] and Chunlin Li [1,3,*]

1   School of Resources and Environmental Engineering, Anhui University, Hefei 230601, China;
    ly3476034225@163.com (Y.L.); lwqldq@163.com (W.L.); wangyutao323@163.com (Y.W.);
    wgy19980914@163.com (G.W.)

2   Forestry Science and Technology Extension Center in Sihong County, Suqian 223900, China;
    suli021777@163.com

3   Anhui Province Key Laboratory of Wetland Ecosystem Protection and Restoration, Anhui University,
    Hefei 230601, China

*   Correspondence: lichunlin@ahu.edu.cn; Tel.: +86-182-2660-1609

**Abstract:** The degradation and loss of natural wetlands has caused severe crises for wetland taxa. Meanwhile, constructed wetlands are expanding significantly and facing dramatic environmental changes. Exploring the responses of wetland organisms, particularly zooplankton, may have important implications for the management of wetlands. Environmental and zooplankton samples were collected from 34 subsidence wetlands created by underground coal mining across the North China Plain in August 2021. We used generalized linear models and redundancy analysis to test zooplankton responses to environmental variables, with the relative importance quantified by variation partitioning. We identified 91 species, divided into 7 functional groups, with the highest density of rotifer filter feeders (RF, 2243.4 ± 499.4 ind./L). Zooplankton species richness was negatively correlated with electrical conductivity (EC), chlorophyll-a, total phosphorus, and pH. The Shannon–Weiner and Pielou evenness indices were positively correlated with transparency and negatively correlated with the photovoltaic panel area (AS). Rotifer predators (RCs) and RF densities were positively correlated with cropland area and dissolved oxygen, but negatively correlated with AS. Small crustacean filter feeders positively correlated with AS, whereas medium crustacean feeders (MCFs) positively correlated with EC. AS was the most critical variable affecting the zooplankton community. Our study showed that the spatial pattern of zooplankton communities was shaped by environmental heterogeneity across the subsidence wetlands, providing implications for the management and conservation of these constructed wetlands.

**Keywords:** subsidence wetland; zooplankton; functional groups; environmental variables; wetland ecosystem

## 1. Introduction

Over two-thirds of natural wetlands have been degraded and lost [1], severely threatening wetland taxa worldwide [2,3]. Constructed wetlands are increasing in number and may provide compensatory habitats for wetland taxa [4–6]. They may play a role in wetland ecosystems and compensate for natural wetlands to a certain extent when these wetlands are well managed. However, the environments of constructed wetlands can change very often in response to humans [7]. Exploring how wetland taxa adapt to drastic environmental changes in constructed wetlands will help us to understand the mechanisms for maintaining the biodiversity and ecosystem services of wetlands.

Among the many wetland taxa, zooplankton, as the link between producers and consumers, play a crucial role in maintaining the health and stability of wetland ecosystems [8–10]. Zooplankton are an important food source for almost all freshwater fish

species and controlling phytoplankton populations through filter feeding has a significant effect on eutrophication status. Changes in zooplankton communities affect the nutrient structure and stability of wetland ecosystems and have become a topic of global concern [11]. Zooplankton are highly dependent on water; therefore, the physical and chemical properties of water may directly affect zooplankton [12]. The physicochemical factors of water usually directly affect the growth, development, and proliferation of zooplankton [13]. These individual-level effects transmitted to the community level may cause changes in the zooplankton community structure [14]. Indirect factors may also alter the zooplankton communities by influencing the physical and chemical indicators of the water [15]. For example, urbanization and agricultural activities transport nutrients to wetlands through surface runoff, accelerating the process of water eutrophication and increasing the population's tolerance to pollution, thus altering the community structure, and mostly negatively correlated with the photovoltaic panel area [16]. Zooplankton are sensitive to living conditions and are often used as biological indicators of environmental change; therefore, they are crucial for wetland management [17,18]. Exploring how zooplankton communities respond to such changes is vital for understanding the maintenance of zooplankton biodiversity in wetlands.

Compared with natural wetlands, dramatic changes in the environment of constructed wetlands may significantly affect wetland taxa; therefore, the response mechanisms of taxa in constructed wetlands deserve attention [3,19]. Environmental factors inside wetlands, such as water depth, topography, and hydrological conditions, are significantly affected by human activities, which are very different from those in natural wetlands [20]. Additionally, human activities in wetlands, such as aquaculture and the laying of photovoltaic panels, have caused changes in the water environment [21]. However, the environment around wetlands, such as land use patterns, is affected by human activities, and more nutrients and pollutants are transported to wetlands through rainfall and surface runoff, resulting in changes in the aquatic environment [22,23]. Investigating wetland taxa, especially zooplankton, in response to the dramatic environmental changes caused by human activities is important for managing constructed wetlands.

Subsidence wetland is a new type of constructed wetland. Formed in the last 30 years and still expanding, this variety of wetland is produced by continuous underground mining activities [24–26]. China is a large coal mining country with abundant coal reserves [27], with more than 81.8% of the coal production coming from underground mining [28]. By 2020, the land subsidence caused by coal mining exceeded $3.5 \times 10^4$ ha [29]. Due to high groundwater levels and abundant rainfall, nearly two-thirds of the subsided area of the North China Plain has been waterlogged [24]. The original terrestrial ecosystem has been transformed into a wetland ecosystem, creating hundreds of independent wetlands of different sizes that may provide critical alternative habitats for wetland organisms [30,31]. Similarly, subsidence created by underground mining has been recorded in other countries, and the resulting wetlands are also important hotspots for research into aquatic biodiversity [32,33]. Biodiversity studies of coalmine subsidence wetlands in the same region in China have mainly focused on single populations, and higher or lower trophic levels, such as comparing phytoplankton levels to the presence of birds [2,3,34]. In contrast, there are few overall studies of zooplankton as the middle position in the food chain. Further research is needed to understand the biodiversity of subsidence wetlands, which has been largely overlooked.

We predicted that high-nutrient wetlands would have higher zooplankton densities, but that communities would have lower alpha diversity because of the high proliferation of a few species [35]. We also predicted that different zooplankton functional groups would respond differently to environmental factors [36]. Additionally, factors associated with human activities may have a higher interpretative variance for the zooplankton community because the environmental factors in constructed wetlands are drastically changed by anthropogenic influences [37]. Given the increasing interference of human activities in the natural environment, the response of zooplankton to environmental change has become an

important topic in research and wetland management. Subsidence wetlands in the North China Plain are expanding under the influence of continuous underground coal mining and play an increasingly important role in regional biodiversity conservation. Therefore, exploring how zooplankton communities in constructed wetlands respond to drastic environmental changes will provide new insights into the maintenance of biodiversity in aquatic communities in human-dominated landscapes and have important implications for the effective management of wetland ecosystems.

## 2. Materials and Methods

### 2.1. Study Area

The Huainan mining area, situated in the southern part of the North China Plain (32.73–33.73° N, 116.03–117.52° E; Figure 1), is one of China's largest coal production bases [24]. The flat terrain and abundant water resources of this region, which is characterized by a warm temperate monsoon climate, are attributed to an average annual precipitation of 970 mm. The landscape is mainly composed of croplands. However, for over a century, massive underground mining has led to large-scale ground deformation and subsidence. By 2020, the extent of land subsidence within the Huainan mining area had expanded to $3.5 \times 10^4$ ha, encompassing a flooded area of $2.6 \times 10^4$ ha [38]. These subsidence wetlands were formed in different years with different sizes and clear boundaries, and many of them continue to expand owing to ongoing underground coal mining activities.

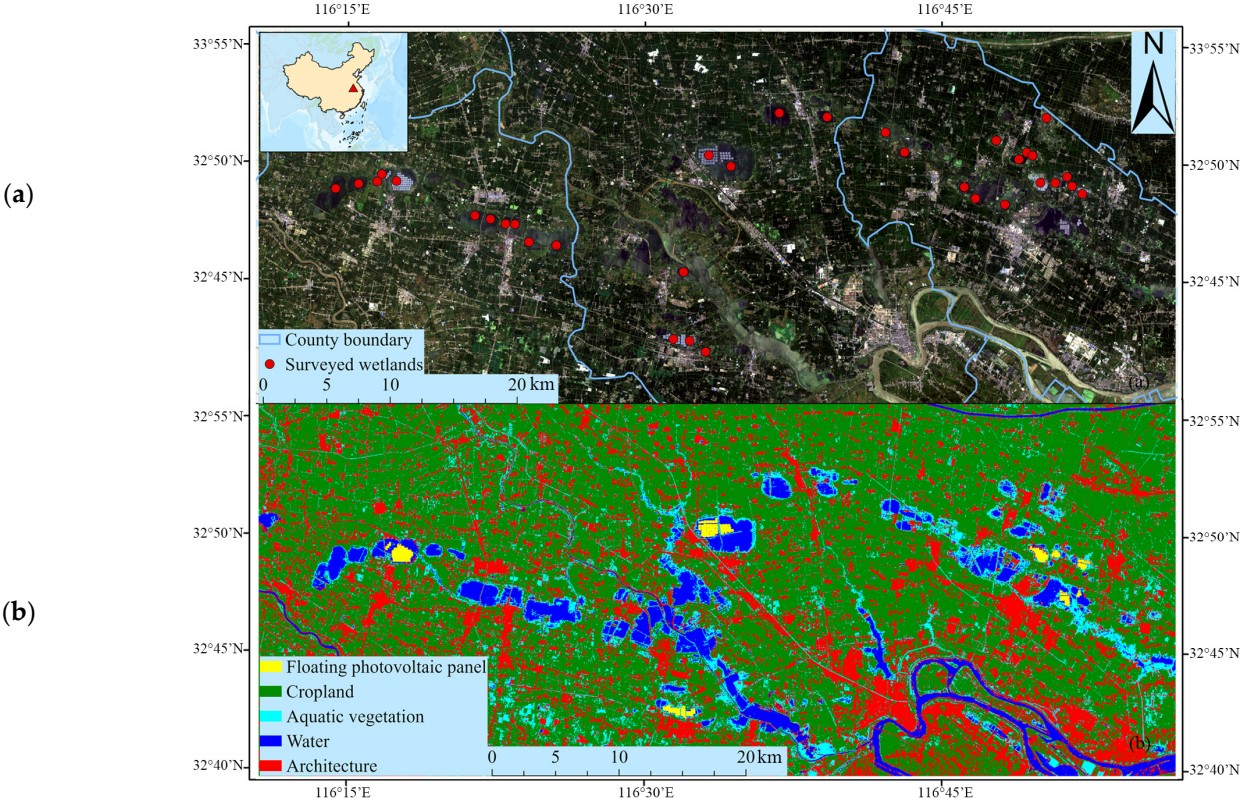

**Figure 1.** The 34 subsidence wetlands (**a**) for sampling zooplankton communities and the land cover map (**b**) of the study area in the Huainan coal mining area, China.

### 2.2. Zooplankton Sampling

In August 2021, zooplankton samples were collected from 34 randomly selected subsidence wetlands in the Huainan coalmine area. Four sampling sites were established in each wetland: two were littoral and two were pelagic. The littoral sites were 5–10 m away from the wetland boundary, and the pelagic sites were located in the center of the wetlands. The distance between adjacent sampling sites was >200 m. Zooplankton samples

were collected vertically from different water layers at each sampling site. When the water depth was less than 10 m, we collected zooplankton samples from 0.5 m below the surface and 0.5 m above the bottom. When the wetland depth was >10 m, another sample was collected from the medium layer to improve its representativeness.

Rotifer samples were collected in 1 L plastic bottles and fixed in 1% Lugol's iodine solution in the field. Each rotifer sample was concentrated to 30 mL after 48 h of precipitation in the laboratory. The crustacean samples were collected with 10 L of mixed water through a 60 μm plankton net, with an opening area of 346 cm$^2$. They were then transferred to 50 mL plankton bottles and preserved with 5% formalin in the field. The zooplankton count was performed using a light microscope (Olympus, BX53; OLYMPUS TOKYO, Tokyo, Japan), and 1 mL sub-samples of rotifers were counted in a counting chamber twice at $100\times$ magnification, while 5 mL sub-samples of crustaceans were counted at $40\times$ magnification for all 50 mL samples. Species identification was conducted as described previously [39–42]. The rotifer density of 1 L water was calculated using an average of two sub-samples, while the density of crustaceans was calculated using the sum of 50 mL samples. The density and alpha diversity indices of the zooplankton were calculated for each sample and averaged for the entire wetland [3].

We divided the collected zooplankton into the following functional groups based on size and feeding habits [10,43,44]: rotifer filter feeders (RFs), rotifer carnivores (RCs), small crustacean filter feeders (SCFs), medium crustacean filter feeders (MCFs), medium crustacean carnivores (MCCs), large crustacean filter feeders (LCFs), and large crustacean carnivores (LCCs; Table 1).

**Table 1.** Description and classification of zooplankton functional group in the 34 subsidence wetlands in the Huainan coal mining area in China.

| Scientific Name | Description | Functional Group | Size (mm) |
|---|---|---|---|
| *Rotaria tardigrada* | Rotifer filter feeders | RFs | |
| *Colurella obtusa* | Rotifer filter feeders | RFs | |
| *Lepadella quinquecostata* | Rotifer filter feeders | RFs | |
| *Brachionus angularis* | Rotifer filter feeders | RFs | |
| *Brachionus calycifiorus* | Rotifer filter feeders | RFs | |
| *Brachionus forficula* | Rotifer filter feeders | RFs | |
| *Brachionus budapestiensis* | Rotifer filter feeders | RFs | |
| *Brachionus capsuliflorus* | Rotifer filter feeders | RFs | |
| *Brachionus urceus* | Rotifer filter feeders | RFs | |
| *Brachionus falcatus* | Rotifer filter feeders | RFs | |
| *Brachionus caudatus* | Rotifer filter feeders | RFs | |
| *Brachionus diversicornis* | Rotifer filter feeders | RFs | |
| *Platyias quadricornis* | Rotifer filter feeders | RFs | |
| *Platyias militaris* | Rotifer filter feeders | RFs | |
| *Anuraeopsis fissa* | Rotifer filter feeders | RFs | |
| *Keratella cochlearis* | Rotifer filter feeders | RFs | |
| *Keratella valga* | Rotifer filter feeders | RFs | |
| *Keratella qudrata* | Rotifer filter feeders | RFs | |
| *Notholca labis* | Rotifer filter feeders | RFs | |
| *Lecane luna* | Rotifer filter feeders | RFs | |
| *Lecane ungulata* | Rotifer filter feeders | RFs | |
| *Lecane pioenensis* | Rotifer filter feeders | RFs | |
| *Lecane eutarsa* | Rotifer filter feeders | RFs | |
| *Lecane closterocerca* | Rotifer filter feeders | RFs | |
| *Lecane ludwigii* | Rotifer filter feeders | RFs | |
| *Lecane curvicornis* | Rotifer filter feeders | RFs | |
| *Monostyla stenroosi* | Rotifer filter feeders | RFs | |
| *Monostyla hamata* | Rotifer filter feeders | RFs | |
| *Monostyla closterocerca* | Rotifer filter feeders | RFs | |
| *Monostyla crenata* | Rotifer filter feeders | RFs | |
| *Monostyla bulla* | Rotifer filter feeders | RFs | |
| *Monostyla elachis* | Rotifer filter feeders | RFs | |
| *Asplanchna priodonta* | Rotifer carnivores | RCs | |
| *Ascomorpha ecaudis* | Rotifer filter feeders | RFs | |
| *Diurella rousseoeti* | Rotifer filter feeders | RFs | |

**Table 1.** *Cont.*

| Scientific Name | Description | Functional Group | Size (mm) |
|---|---|---|---|
| *Diurella stylata* | Rotifer filter feeders | RFs | |
| *Diurella dixon-nuttalli* | Rotifer filter feeders | RFs | |
| *Diurella collaris* | Rotifer filter feeders | RFs | |
| *Trichocerca cylindrica* | Rotifer filter feeders | RFs | |
| *Trichocerca capucina* | Rotifer filter feeders | RFs | |
| *Trichocerca pusilla* | Rotifer filter feeders | RFs | |
| *Trichocerca lophoessa* | Rotifer carnivores | RCs | |
| *Trichocerca elongata* | Rotifer filter feeders | RFs | |
| *Synchaeta pectinata* | Rotifer filter feeders | RFs | |
| *Polyarthra euryptera* | Rotifer filter feeders | RFs | |
| *Polyarthra trigla* | Rotifer carnivores | RCs | |
| *Polyarthra vulgaris* | Rotifer filter feeders | RFs | |
| *Mytilina ventralis* | Rotifer filter feeders | RFs | |
| *Pompholyx complanata* | Rotifer filter feeders | RFs | |
| *Pedalia mira* | Rotifer filter feeders | RFs | |
| *Filinia minuta* | Rotifer filter feeders | RFs | |
| *Filinia terminalis* | Rotifer filter feeders | RFs | |
| *Filinia opoliensis* | Rotifer filter feeders | RFs | |
| *nauplius* | Small crustacean filter feeders | SCFs | <0.70 |
| Sinocalanus Burckhardt | Large crustacean filter feeders | LCFs | >1.50 |
| *Schmackeria inopinus* | Medium crustacean feeders | MCFs | 0.70–1.50 |
| *Schmackeria forbesi* | Medium crustacean feeders | MCFs | 0.70–1.50 |
| *Heliodiaptomus serratus* | Medium crustacean feeders | MCFs | 0.70–1.50 |
| *Sinodiaptomus sarsi* | Large crustacean filter feeders | LCFs | >1.50 |
| *Neodiaptomus schmackeri* | Medium crustacean feeders | MCFs | 0.70–1.50 |
| *Eodiaptomus sinensis* | Medium crustacean feeders | MCFs | 0.7–1.5 |
| *Onychocamptus mohammed* | Small crustacean filter feeders | SCFs | <0.70 |
| *Limnoithona sinensis* | Small crustacean filter feeders | SCFs | <0.70 |
| *Macrocyclops albidus* | Medium crustacean carnivores | MCCs | 0.70–1.50 |
| *Macrocyclops distinctus* | Medium crustacean feeders | MCFs | 0.70–1.50 |
| *Eucylops serrulatus* | Medium crustacean feeders | MCFs | 0.70–1.50 |
| *Microcyclops varicans* | Medium crustacean feeders | MCFs | 0.70–1.50 |
| *Mesocyclops leuckarti* | Medium crustacean carnivores | MCCs | 0.70–1.50 |
| *Thermocyclops hyalinus* | Medium crustacean carnivores | MCCs | 0.70–1.50 |
| *Cyclops strenuus* | Large crustacean carnivores | LCCs | >1.50 |
| *Leptodora kindti* | Large crustacean carnivores | LCCs | >1.50 |
| *Diaphanosoma leuchtenbergianum* | Medium crustacean feeders | MCFs | 0.70–1.50 |
| *Diaphanosoma brachyurum* | Medium crustacean feeders | MCFs | 0.70–1.50 |
| *Diaphanosoma sarsi* | Medium crustacean feeders | MCFs | 0.70–1.50 |
| *Diaphanosoma excisum* | Medium crustacean feeders | MCFs | 0.70–1.50 |
| *Daphnia pulex* | Large crustacean filter feeders | LCFs | >1.50 |
| *Daphnia hyalina* | Large crustacean filter feeders | LCFs | >1.50 |
| *Daphnia cucullata* | Large crustacean filter feeders | LCFs | >1.50 |
| *Ceriodaphnia pulchella* | Small crustacean filter feeders | SCFs | <0.70 |
| *Ceriodaphnia cornuta* | Small crustacean filter feeders | SCFs | <0.70 |
| *Ceriodaphnia quadrangula* | Small crustacean filter feeders | SCFs | <0.70 |
| *Scapholeberis mucronata* | Medium crustacean feeders | MCFs | 0.70–1.50 |
| *Moina micrura* | Small crustacean filter feeders | SCFs | <0.70 |
| *Moina rectirostris* | Medium crustacean feeders | MCFs | 0.70–1.50 |
| *Bosmina longirostris* | Small crustacean filter feeders | SCFs | <0.70 |
| *Bosmina fatalis* | Small crustacean filter feeders | SCFs | <0.70 |
| *Bosmina coregoni* | Small crustacean filter feeders | SCFs | <0.70 |
| *Bosminopsis Richard* | Small crustacean filter feeders | SCFs | <0.70 |
| *Alona guttata* | Small crustacean filter feeders | SCFs | <0.70 |
| *Chydorus sphaericus* | Small crustacean filter feeders | SCFs | <0.70 |
| *Pleuroxus hamulatus* | Small crustacean filter feeders | SCFs | <0.70 |

### 2.3. Habitat Variable

We quantified 13 environmental variables that could affect zooplankton communities, including physicochemical and anthropogenic disturbances (Table 2). Two 1 L water samples were collected in opaque plastic bottles, stored at low temperatures, and transported to the laboratory. During the field surveys, pH, dissolved oxygen (DO), and electrical conductivity (EC) were measured using a Hach HQ40d portable multimeter; transparency (SD) and water depth were measured using a Secchi disk. In the laboratory, total phospho-

rus (TP, 0.5 L sample), total nitrogen (TN, 0.5 L sample), and chlorophyll-a (Chl-a, 0.5 L sample) were measured using standard analytical methods [45]. For the convenience of the reviewers and readers, please refer to Li [3] for the specific sampling and experimental methods employed. Other variables were obtained from the land cover map. To obtain a land cover map of the study area, we downloaded remotely sensed images without cloud cover from the US Geological Survey website on 1 August 2021 http://glovis.usgs.gov (accessed on 10 June 2022). The obtained remote sensing images were radiometrically and geometrically (systematically) corrected using ground control points and ephemeris data in ENVI5.3. They were then re-projected onto zone 50 (north) of the Universal Transverse Mercator Projection 1984 coordinate system. The study areas were subjected to supervised classification using the maximum likelihood classification method. Five land cover types were identified: water, aquatic vegetation, cropland, architecture, and floating photovoltaic panels (Figure 1). To verify the classification, training samples were used, and the overall accuracy was determined to be 96.85% with a κ coefficient of 0.954, indicating a high classification accuracy.

**Table 2.** Habitat variables of zooplankton community structure in the 34 subsidence wetlands in the Huainan coal mining area in China.

| Habitat Variables | Description | Range | Mean | SE |
|---|---|---|---|---|
| pH | pH | 7.15–9.00 | 7.98 | 0.08 |
| WD (m) | Water depth | 2.10–15.10 | 6.50 | 0.50 |
| DO (mg/L) | Dissolved oxygen | 4.00–13.85 | 7.26 | 0.38 |
| EC (us/cm) | Electric conductivity | 449.37–1788.63 | 764.80 | 44.14 |
| SD (m) | Transparency | 0.23–14.8 | 6.10 | 0.48 |
| TP (mg/L) | Total phosphorus concentration | 0.06–1.22 | 0.34 | 0.04 |
| TN (mg/L) | Total nitrogen concentration | 0.26–2.2 | 1.12 | 0.34 |
| Chl-a (μg/L) | Chlorophyll-a concentration | 730.24–813.85 | 281.54 | 32.61 |
| AW (km$^2$) | Area of each wetland | 0.04–3.91 | 1.09 | 0.17 |
| AA (km$^2$) | Area of aquatic vegetation in each wetland | 0.01–0.30 | 0.09 | 0.01 |
| AS (km$^2$) | Area of floating photovoltaic panel in each wetland | 0.16–1.73 | 0.64 | 0.21 |
| AC (km$^2$) | Area of cropland in each wetland within a 2 km buffer zone | 6.70–17.22 | 10.33 | 0.43 |
| AD (km$^2$) | Area of architecture in each wetland within a 2 km buffer zone | 2.24–7.53 | 4.41 | 0.25 |

### 2.4. Statistical Analyses

We used a generalized linear model to analyze the relationships between zooplankton species richness, Pielou evenness index, the Shannon–Wiener diversity index, and environmental factors. First, we calculated the variance inflation factor (VIF) and removed TN (VIF > 10). For species richness, a GLM with a negative binomial distribution was used, whereas the Pielou evenness and Shannon–Wiener indices were transformed using square-root methods and analyzed using a GLM with a Gaussian distribution. A backward selection procedure was used to select the final model. We used the indirect ordination method to analyze differences in the composition of the functional groups of zooplankton communities and their relationships with environmental factors. We conducted a detrended correspondence procedure for the zooplankton communities. As the axis length was 0.77, we selected an RDA model. Before conducting RDA, TN was excluded from the model because of collinearity. To satisfy the multivariate normality hypothesis, we transformed all environmental variables and zooplankton functional group density data, except for pH and architectural area, by log10 (X + 1). A 999-permutation Monte Carlo permutation test was used to examine the significance of the variance in the RDA gradient. The significant differences between the densities of each functional group were assessed using Kruskal–Wallis testing. The relative importance of each variable was determined via

variation partitioning using the adjusted R-squared method in the RDA. For each species, we calculated the McNaughton dominance index (Y), $Y = (N_i/N) \times f_i$, where $N_i$ was the total number of individuals species i in all samples, N was the total number of all species in all samples, and $f_i$ was the occurrence frequency of species i. When Y > 0.02, the species was registered as the dominant species [46]. All statistical analyses were performed using the "vegan" packages in R 3.4.1.

## 3. Results

### 3.1. Habitat Variables

The sampled wetland was weakly alkaline, and the average water depth was 6.50 m, with pH and DO ranges of 7.1–9 and 4–13.9 mg/L, respectively. The nutrient state of the wetland was indicated by TN, TP, and chlorophyll-a, with respective values of 1.12 (±0.34) mg/L, 0.34 (±0.04) mg/L, and 2.82 (±0.31) μg/L. The EC and SD in wetlands varied significantly, standing at 764.80 (±43.48) us/cm and 6.10 (±0.48) m, respectively. The sizes of wetlands ranged from 0.04 to 3.9 km², and the cover of aquatic vegetation had an average area of 0.09 (±0.01) km². The floating photovoltaic panels were fitted with seven wetlands, covering an area of 0.27 to 1.7 km². Within the 2 km buffer zone surrounding each wetland, the land cover types were cropland and architecture area, and mean areas were 10.33 (±0.43) km² and 4.41(±0.25) km² (Table 2).

### 3.2. Composition of Zooplankton Community

We recorded 91 zooplankton species from 22 families and 46 genera and divided them into seven functional groups (LCCs, LCFs, MCCs, MCFs, RCs, RFs, RCs, and SCFs; Table 1). The species richness was the highest in the rotifer filter feeders (10.3 ± 0.5), followed by MCFs (5.0 ± 0.3) and SCFs (4.9 ± 0.4). The significant differences between the densities of the different functional groups ($p < 0.05$) and the dominant species were as follows: *Polyarthra trigla* (24.53%; Y = 0.25), *Trichocerca pusilla* (15.28%; Y = 0.23), *Anuraeopsis fissa* (1218%; Y = 1.12), and nauplius (1.73%, Y = 0.02; Table 3). The density was the highest in the RFs (Figure 2), with an average density of 2243.4 ± 499.4 ind./L, which was the contribution of *Brachionus forficula*, *Brachionus angularis*, *Trichocerca pusilla*, *Trichocerca capucina*, etc. The following are the data for RCs with an average density of 787.3 ± 124.1 ind./L, which was the contribution of *Polyarthra trigla* (Figure 2). The density of MCFs was 146.5 ± 14.2 ind./L, which was the contribution of *Eucylops serrulatus*, *Microcyclops varicans,* and *Diaphanosoma sarsi*. The dominant species of LCF were *Sinodiaptomus sarsi*, *Daphnia pulex,* and *Daphnia cucullate*, while the dominant species of MCCs were *Thermocyclops hyalinus* and *Mesocyclops leuckarti*. The dominant species in SCFs and LCCs were nauplius and *Cyclops strenuous*, respectively.

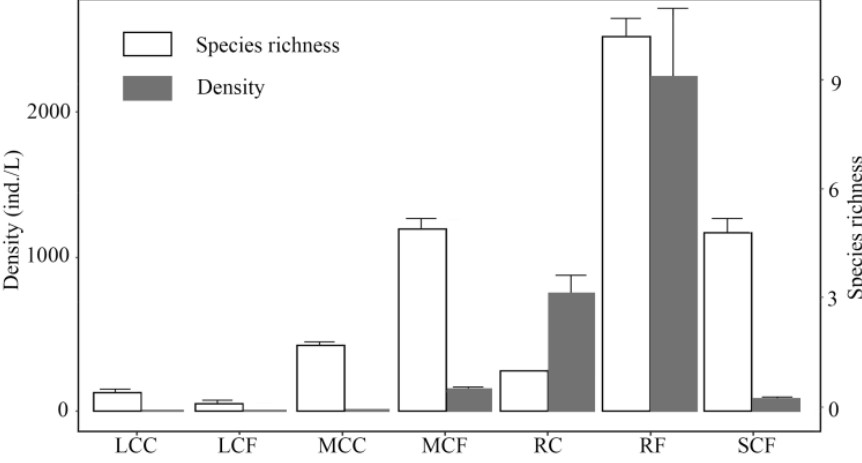

**Figure 2.** Density and species richness of functional groups of zooplankton in the 34 subsidence wetlands in the Huainan coal mining area in China.

**Table 3.** Results of relative abundance analysis of the four dominant species to determine the top contributions to abundance-based community structure in the subsidence wetlands in the Huainan coal mining subsidence area, China.

| Scientific Name | Relative Abundance (%) | McNaughton Dominance Index (Y) |
|---|---|---|
| Polyarthra trigla | 24.53 | 0.25 |
| Trichocerca pusilla | 15.28 | 0.23 |
| Anuraeopsis fissa and | 12.18 | 0.12 |
| nauplius | 10.73 | 0.02 |

### 3.3. Effects of Environmental Variables on Zooplankton Community Diversity

Species richness was negatively correlated with TP concentration, Chl-a concentration, EC, and pH. The Shannon–Weiner diversity and Pielou evenness indices were positively correlated with SD, but negatively correlated with the floating photovoltaic panel area (AS). Additionally, the Shannon–Weiner diversity index was negatively correlated with the TP concentration (Table 4).

**Table 4.** Summary of generalized linear model results of zooplankton diversity index and able in the 34 subsidence wetlands in the Huainan coal mining area, China.

| Diversity Index | Environment Variable | Coefficient | *p* |
|---|---|---|---|
| Species richness | pH | −0.14 | 0.01 |
| | Conductivity | −0.0002 | <0.05 |
| | Total phosphorus concentration | −0.30 | <0.05 |
| | Chlorophyll-a concentration | −0.04 | <0.05 |
| Pielou evenness index | Transparency | 0.001 | <0.05 |
| | Area of floating photovoltaic panel in each wetland | −0.10 | 0.02 |
| Shannon–Weiner diversity index | Transparency | 0.02 | 0.05 |
| | Total phosphorus concentration | −1.45 | 0.04 |
| | Area of floating photovoltaic panel in each wetland | −1.34 | 0.03 |

### 3.4. Correlation between Functional Groups of Zooplankton and Habitat Variables

In the final RDA model, the four variables exhibited a significant impact, as determined by the Monte Carlo test (Table 5). The first two RDA axes collectively elucidated 87.1% of the variance within the zooplankton community, with respective eigenvalues of 0.41 and 0.07 (Table 5). The densities of RFs and RCs were positively correlated with cropland area (AC) and DO and negatively correlated with the AS. The density of MCFs was positively correlated with AC and EC and negatively correlated with DO. The density of small crustacean feeders was positively correlated with the AS but negatively correlated with DO and AC. Given the limited projection magnitude, the interplay between large crustacean feeders, large crustacean carnivora, and medium crustacean carnivora group densities and environmental factors was not analyzed (Figure 3). Variation partitioning showed that the variation in the zooplankton communities was mainly explained by the AS (34.0%), followed by electric conductivity (10.3%), DO (7.3%), and AC (7.1%) (Figure 4).

**Table 5.** Summary of RDA models of the relationships between zooplankton communities and habitat variables in the 34 subsidence wetlands in the coal mining area in China.

| Information | Numerical Value |
|---|---|
| Axis length | 0.79 |
| Significant variables in RDA model | AS ($p < 0.05$) |
| | EC ($p < 0.05$) |

**Table 5.** *Cont.*

| Information | Numerical Value |
|---|---|
| | AC ($p < 0.05$) |
| | DO ($p < 0.05$) |
| Proportion of total variance explained | 44.95% |
| Constrained eigenvalue of RDA 1 | 0.41 |
| Constrained eigenvalue of RDA 2 | 0.07 |
| Proportion of constrained variance explained by RDA 1 | 74.05% |
| Proportion of constrained variance explained by RDA 2 | 13.01% |
| Cumulative constrained variance explained | 87.06% |
| Model significance by Monte Carlo test | F = 7.74, $p < 0.05$ |

AS, area of floating photovoltaic panel in each wetland; AC, area of cropland in each wetland within 2 km buffer zone; EC, electrical conductivity; DO, dissolved oxygen.

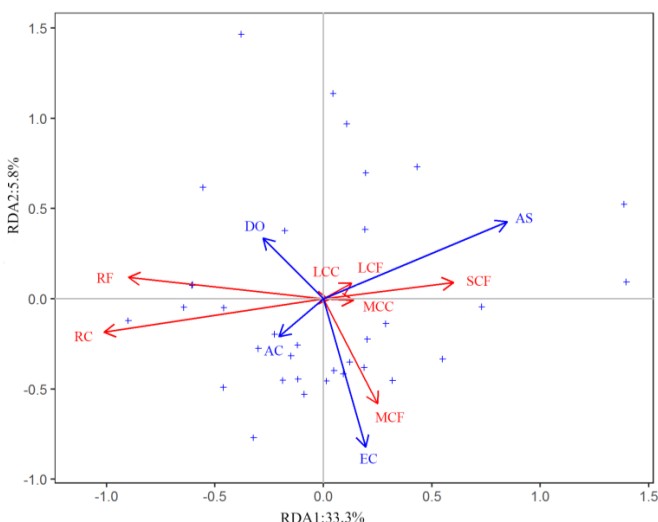

**Figure 3.** RDA of zooplankton functional group density and habitat variables, the 34 subsidence wetlands in the Huainan coal mining area, China. ($p < 0.05$). AS, area of floating photovoltaic panel in each wetland; AC, area of cropland in each wetland within 2 km buffer zone; EC, electric conductivity; DO, dissolved oxygen.

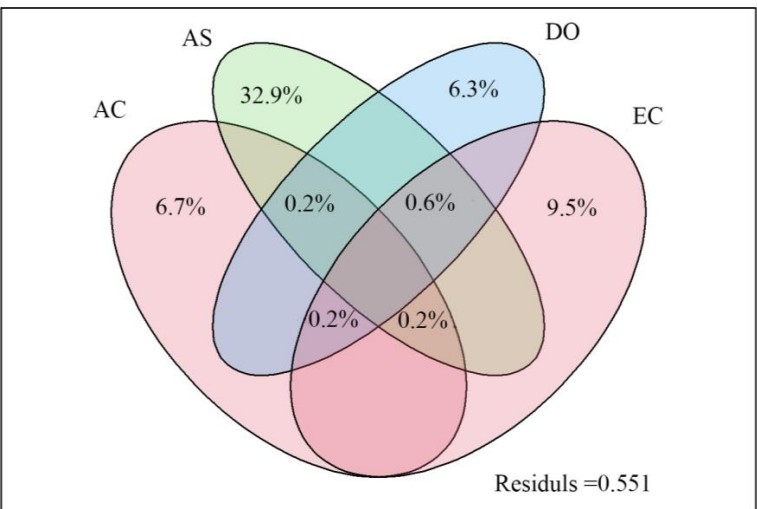

**Figure 4.** Variation partitioning of the variance of zooplankton communities in the 34 subsidence wetlands in the Huainan coal mining area, China. Value < 0 not shown. AS, area of floating photovoltaic panel in each wetland; AC, area of cropland in each wetland within 2 km buffer zone; EC, electric conductivity; DO, dissolved oxygen.

## 4. Discussion

We found that all the subsidence wetlands were weakly alkaline, which was similar to other wetlands in the same region [47,48]. The wetlands had high concentrations of nitrogen and phosphorus, which was similar to other subsidence wetlands in the North China Plain [49]. The regional climate is a warm temperate monsoon climate and the annual rainfall reaches 970 mm. Compared with other artificial wetlands in the same region, we recorded a relatively higher number of zooplankton species [47]. Moreover, we found similar species compositions in other single-subsidence wetlands in the North China Plain [48,49], where small species (e.g., rotifer filter feeders and rotifer carnivores) had significantly higher densities than other taxa and dominated the community. Specifically, there were significantly more species of rotifer filter feeders were and they had higher densities than other functional groups in the community. However, zooplankton are highly sensitive to changes in environmental factors [50].

As predicted, high-nutrient wetlands have higher zooplankton densities, but their communities have lower species richness, Shannon–Weiner scores, and evenness indices. Our results showed that eutrophication occurs to different degrees in subsidence wetlands [51,52]. Wetlands with high nutrient levels have higher zooplankton densities, mainly because of the contribution of the tolerant species *Trichocerca pusilla*, *Anuraeopsis fissaand* and *Brachionus* [53]. At the same time, the large proliferation of these species has limited the growth of other populations, reducing the overall species richness [54,55]. Additionally, higher conductivity and pH environments lead to lower zooplankton species richness because most single-celled zooplankton are intolerant to high conductivity and pH environments owing to osmotic regulation [56]. Notably, Shannon–Weiner diversity and Pielou evenness indices were positively correlated with SD and negatively correlated with AS. Higher SD has been shown to reduce the proliferation of dominant algae in eutrophic environments and to promote the recovery of underwater aquatic plants [57], thereby enhancing habitat heterogeneity, improving water quality, and ultimately contributing to the diversification of zooplankton communities [58]. The introduction of photovoltaic panels suppresses the excessive dominance of a few species [3], but reduces the overall zooplankton diversity by limiting phytoplankton production [59].

Each zooplankton functional group responded differently to environmental factors. RFs (*Brachionus forficula*, *Brachionus angularis*, etc.) and RCs (*Polyarthra trigla*) were positively correlated with AC and DO, but negatively correlated with the AS. Agricultural activities transport large nutrient loads and organic debris to wetlands, which gives RFs and RC, as R-strategy animals, an advantage in their competition with other groups and thus allows them to proliferate [60,61]. Floating photovoltaic panels effectively reduce the availability of light and wind disturbances [62], thereby reducing the excessive advantage of RFs and RCs. SCFs were mainly composed of nauplius. This positively correlated with AS, in contrast to RFs and RCs. Interspecies competition is evident between small crustacean and rotifer filter feeder groups for available resources and habitats [63]. SCFs have low filter-feeding efficiency and reproduction rates and are at a disadvantage in competition with RFs [64]. Floating photovoltaic panels reduce the competitive advantage of RFs and promote their proliferation. So, we found higher levels of small crustacean filter density in wetlands with floating photovoltaic panels. *Eucylops serrulatus*, *Microcyclops varicans*, and *Diaphanosoma sarsi* were the dominant species of MCF, and were positively correlated with conductivity because they have a wider tolerance to conductivity and require more calcium during growth than other taxa [65,66].

Among the many types of environmental factors, human disturbance significantly affects zooplankton communities in subsidence wetlands. Variation partitioning emphasized the significant role played by human disturbances inside and outside wetlands in the formation of zooplankton community structures. Floating photovoltaic panels and agricultural activities in wetlands directly or indirectly change the physical and chemical environment of the water and affect the structure of the zooplankton community [3,67]. In constructed wetlands, such as coal mining subsidence wetlands, it is essential to study

how zooplankton adapt to drastic environmental changes [23,68]. In addition to the above environmental factors, other environmental factors can affect the zooplankton community structure, such as water temperature and hydrological conditions. However, we did not examine the relationships between these environmental factors and zooplankton in this present study. The subsidence wetlands in the North China Plain are located at the same latitude, and the daily fluctuations in water temperature during continuous sampling cannot truly reflect the relationship between water temperature and zooplankton in different wetlands. Therefore, it is necessary to understand how environmental factors such as water temperature affect zooplankton communities, and put more effort into these aspects in future research. Hence, it is essential to comprehend the impact of environmental factors, such as water temperature, on zooplankton communities. We should dedicate further research efforts to this in the future.

### 5. Conclusions

We found abundant zooplankton species in subsidence wetlands due to underground coal mining in the North China Plain, and the rotifer filter feeders had the highest species richness and density. High-nutrient wetlands had higher zooplankton densities because of the proliferation of a few tolerant species, resulting in the lower species richness of zooplankton communities. Additionally, higher transparency promoted the restoration of aquatic vegetation and enhanced habitat heterogeneity, resulting in a higher level of zooplankton diversity index. The introduction of photovoltaic panels effectively reduced the amount of light and phytoplankton content, resulting in low overall zooplankton diversity. Rotifer filter feeders, rotifers carnivora, and small crustacean filter feeders responded in opposite ways to environmental factors because of interspecific competition. Rotifer filter feeders and rotifers carnivora preferred habitats with high nutrition and rich food resources, whereas floating photovoltaic panels reduced the competitive advantage of the former and made small crustacean filter feeders the dominant species. The main human disturbance in the area of floating photovoltaic panels significantly affected the zooplankton community. We predict that zooplankton communities in subsidence wetlands will gradually miniaturize due to the impacts of continued underground coal mining and human interference. Our study provides significant insights into the mechanisms governing the establishment and maintenance of zooplankton community biodiversity in a dramatically changing environment and has substantial implications for the effective management and conservation of constructed wetlands.

**Author Contributions:** All authors contributed to the study conception and design. C.L. conceived the study. Y.L., J.H., W.L., G.W. and Y.W. collected the data. Y.L. and C.L. performed the analyses. Y.L. wrote the first draft of the paper. C.L. revised the manuscript. All authors have read and agreed to the published version of the manuscript.

**Funding:** This work was supported by the National Natural Science Foundation of China (grant 31970500 and 31770571), the Excellent Youth Project of the Anhui Natural Science Foundation (grant 2108085Y09).

**Institutional Review Board Statement:** Not applicable.

**Data Availability Statement:** The data presented in this study are available on request from corresponding author.

**Acknowledgments:** We thank Weiqiang Li and Guangyao Wang for their help and support in the experiment. We would like to thank the supporter of this project.

**Conflicts of Interest:** The authors declare no conflicts of interest.

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
