# Peer review of "How Do Zooplankton Communities Respond to Environmental Factors across the Subsidence Wetlands Created by Underground Coal Mining in the North China Plain?"

_diversity, doi:10.3390/d16050304_

Round 1

Reviewer 1 Report

Comments and Suggestions for Authors

Wetland researches are interesting and actual. However, the results are poorly presented. I recommend that you revise your materials according to my comments.

Author Response

Comments 1: [This paragraph is very vague. Preferably provide data on the aquatic biota diversity. Do similar works exist in China and other countries? Similarities and differences. Have studies been done before in China and/or on surveyed territory? General phrases are presented; it is preferably to provide more complete and specific data. ]

Response 1: [Thank for your professional suggestions, which have significantly improved our manuscript. We have made responding changes to the Introduction section. Similarly, subsidence created by underground mining has been recorded in other countries, and the resulting wetlands are also important hotspots for aquatic biodiversity research [1,2]. Biodiversity studies of coal mine subsidence wetlands in the same region in China have mainly focused on single population, higher or lower trophic levels, such as phytoplankton versus birds [3,4,5]. In contrast, there are few overall studies of zoo-plankton as the middle position in the food chain. Further research is needed to understand the biodiversity of subsidence wetlands, which has been largely overlooked. This change can be found – page 2, line 82-89.

Thanks again for your careful review and constructive suggestions. We have made corresponding modifications according to your comments and those of the other two reviewers, and we hope that our revised manuscript may be approved by you.

1. Pociecha, A.; Wojtal, A. Z.; Szarek-Gwiazda, E.; Cieplok, A.; Ciszewski, D.; Kownacki, A. Response of Cladocera fauna to heavy metal pollution, based on sediments from subsidence ponds downstream of a mine discharge (S. Poland). Water, 2019, 11, 810.

2. Li, C.; Yang, S.; Zha, D.; Zhang, Y.; de Boer, W.F. Waterbird Communities in Subsidence Wetlands Created by Underground Coal Mining in China: Effects of Multi-Scale Environmental and Anthropogenic Variables. Environ Conserv. 2018, 46, 67-75.

3. Li, W.; Wang, Y.; Wang, G.; Liang, Y.; Li, C.; Svenning, J.C. How do rotifer communities respond to floating photovoltaic systems in the subsidence wetlands created by underground coal mining in China? JEM. 2023, 339, 117816.

4. PÄ™czuÅ‚a, W.; Szczurowska, A.; Poniewozik, M. Phytoplankton Community in Early Stages of Reservoir Development--a Case Study from the Newly Formed, Colored, and Episodic Lake of Mining-Subsidence Genesis. Pol J Environ Stud, 2014, 23, 585-591.

5. Jiang, L.; Yao, Y.; Zhang, S.; Wan, L.; Zhou, Z. Effects of Stream Connectivity on Phytoplankton Diversity and Community Structure in Sunken Lakes: A Case Study from an August Survey. Diversity, 2023, 15, 291.]

Comments 2: [Line 86-93 – Delete. This text part is unnecessary. It is Methods.]

Response 2: [Thank for your professional suggestions, which have significantly improved our manuscript. We have made responding changes to the Introduction section.] 

Comments 3: [Line 95 – Species richness, Shannon–Weiner diversity, and evenness’ can be combined with the phrase alpha diversity.]

Response 3: [Thank for your professional suggestions, which have significantly improved our manuscript. We have made responding changes to the Introduction section. This change can be found – page 2, line 91.]

Comments 4: [Figure 1b – KM change to km]

Response 4: [Thank for your professional suggestions, which have significantly improved our manuscript. We have made responding changes to the Methods section, Figure 1. This change can be found – page 3, line 117.]

Comments 5: [Line 119 – Sampling was carried out only once. This is very little to answer the question asked in the manuscript (How do zooplankton communities respond to environmental factors across the subsidence wetlands?). It would be more interesting and correct to see summer data (not only August) or research over several years.]

Response 5: [Sorry for the confusion. First of all, thank you very much for taking the valuable time to review. We would like to explain that we focused our sampling in the summer because many zooplankton grow and reproduce in the summer. In the current study, although the samples were collected only in summer, the number of surveyed wetlands in the analyses was large enough, 34 Subsidence wetlands. Furthermore, four sampling sites were established in each wetland: two littoral and two pelagic. The littoral sites were 5-10 meters away the wetland boundary, and the pelagic sites were located in the center of the wetlands. The distance between adjacent sampling sites was > 200 meters. Zooplankton samples were collected vertically from different water layers at each sampling site. When the water depth was less than 10 meters, we collected zoo-plankton samples from 0.5 meters below the surface and 0.5 meters above the bottom. When the wetland depth was > 10 meters, another sample was collected from the medium layer to improve its representativeness. This change can be found – page 3, line 117-124. We acknowledge that the results may differ between summer and over several years date, and comparing this data may generate some interesting findings. Therefore, we will collect samples in other seasons in the near future and analyze the seasonal differences.

Thanks again for your careful review and constructive suggestions. We have made corresponding modifications according to your comments and those of the other two reviewers, and we hope that our revised manuscript may be approved by you.]

Comments 6: [121, 123 and further – meters or m. Bring about uniformity.]

Response 6: [Thank for your professional suggestions, which have significantly improved our manuscript. We have made responding changes to the Methods section 2.2. This change can be found – page 3, line126; page 4, line128-129.]

Comments 7: [129, 130 – h or hours. 48 h – reiteration.]

Response 7: [Thank for your professional suggestions, which have significantly improved our manuscript. We have made responding changes to the Methods 2.2 section. This change can be found – page 4, line 132-133.]

Comments 8: [123 – Did the distance (200 m) between the points depend on the area of the wetland?]

Response 8: [Sorry for the confusion. First of all, thank you very much for taking the valuable time to review. The distance (200 m) between the points depends on the area of the wetland. In our study, the sizes of wetlands ranged from 0.04 to 3.9 km2, the average of wetlands is 1.09, which is sufficient for our sampling requirements.

Thanks again for your careful review and constructive suggestions. We have made corresponding modifications according to your comments and those of the other two reviewers, and we hope that our revised manuscript may be approved by you.]

Comments 9: [124-127 – Preferably provide data on total depth (to Table 1 too). How did you measure depth (water layer)? What determines the choice of water layers (the medium, the bottom, and the bottom layers). Why was sample collection not carried out in the photic layer? Moreover, water transparency was measured. What device was used to take samples from different depths?]

Response 9: [Thank for your professional suggestions, which have significantly improved our manuscript. We have made responding changes to the Result section 3.1, Table 2. This change can be found – page 4, line 149, 154; page 5-6 , line 207. We used the Secchi disk to measure the water depth (WD/m) and transparency (SD/m) of each wetland. The difference in the vertical distribution of zooplankton in each wetland and the water depth determines the location of the different water layers. Zooplankton samples were collected vertically from different water layers at each sampling site rather than the photic layer. When the water depth was less than 10 meters, we collected zooplankton samples from 0.5 meters below the surface and 0.5 meters above the bottom. When the wetland depth was > 10 meters, another sample was collected from the medium layer to improve its representativeness. This change can be found – page 4, line 126; page 5, 127-130.

Thanks again for your careful review and constructive suggestions. We have made corresponding modifications according to your comments and those of the other two reviewers, and we hope that our revised manuscript may be approved by you.]

Comments 10: [136 – “diversity indices”. Indicate which ones and add references.]

Response 10: [Thank for your professional suggestions, which have significantly improved our manuscript. We have made responding changes to the Methods section 2.2. This change can be found – page 4, line 141-142.]

Comments 11: [138-142 – It is not clear why and for what purpose the authors characterized zooplankton by functional groups. Biomass, like the abundance of zooplankton, is an informative and important characteristic. Moreover, the authors presented groups of organisms with different sizes. Preferably provide.]

Response 11: [Sorry for the confusion. First of all, thank you very much for taking the valuable time to review. We divided zooplankton into 8 functional groups based on their size and feeding patterns to better understand the response of functional groups to environmental factors. We acknowledge that biomass and density of zooplankton are among the important information characteristics, but we only use density because the community contribution of some zooplankton species with small biomass (such as rotifer) may be overlooked. We have found examples of density only used in other studies [1-5]. The groups of organisms with different sizes can be found in Table 3. This change can be found – page 6, line 234.

Thanks again for your careful review and constructive suggestions. We have made corresponding modifications according to your comments and those of the other two reviewers, and we hope that our revised manuscript may be approved by you.

1. Gomes, L. F.; Barbosa, J. C.; de Oliveira Barbosa, H.; Vieira, M. C.; Vieira, L. C. G.  Environmental and spatial influences on stream zooplankton communities of the Brazilian Cerrado. Community Ecol. 2020, 21(1), 25-31.

2. Li, W.; Wang, Y.; Wang, G.; Liang, Y.; Li, C.; Svenning, J.C. How do rotifer communities respond to floating photovoltaic systems in the subsidence wetlands created by underground coal mining in China? JEM. 2023, 339, 117816.

3. Zhao, K.; Wang, L.; You, Q.; Pan, Y.; Liu, T.; Zhou, Y. Influence of cyanobacterial blooms and environmental variation on zooplankton and eukaryotic phytoplankton in a large, shallow, eutrophic lake in China. Sci Total Environ. 2021, 773, 145421.

4. Krztoń, W., & Kosiba, J. Variations in zooplankton functional groups density in freshwater ecosystems exposed to cyanobacterial blooms. Sci Total Environ. 2020, 730, 139044.

5. Mondal, S.; Palit, D.; & Hazra, N. Study on composition and spatio-temporal variation of zooplankton community in coal mine generated pit lakes, West Bengal, India. Tropical Ecology.  2023, 64(2), 352-368.]

Comments 12: [192-194 and further – km2 instead km2.]

Response 12: [Thank for your professional suggestions, which have significantly improved our manuscript. We have made responding changes to the Results section 3.1. This change can be found – page 5, line 201-205.]

Comments 13: [Tables and throughout the manuscript – Unify the way of presenting numerical data. Try, as much as possible, to use the same number of decimal places after the decimal dot (one or two).]

Response 13: [Thank for your professional suggestions, which have significantly improved our manuscript. We have made responding changes to the Results section. This change can be found – page 5-6, Table 1, line 207; page 6, Table 2, line 230; page 8-9, Table 3, line 234; page 9, Table 4, line 244; page 10, Table 5, line 262.]

Comments 14: [200 – Which species dominated? Preferably provide.]

Response 14: [Thank for your professional suggestions, which have significantly improved our manuscript. We have made responding changes to the Results section. This change can be found – page 6, line 214-225.]

Comments 15: [Figure 3 - It would be interesting which wetlands diverged on the graph. Preferably provide.]

Response 15: [Sorry for the confusion. First of all, thank you very much for taking the valuable time to review. Figure 3 provides information on environmental factors, zooplankton functional groups, and each subsidence wetland, with blue representing environmental factors, red representing functional groups, and gray plus signs representing wetlands.

Thanks again for your careful review and constructive suggestions. We have made corresponding modifications according to your comments and those of the other two reviewers, and we hope that our revised manuscript may be approved by you.]

Comments 16: [254-257 –Its general phrase, be specific.]

Response 16: [Thank for your professional suggestions, which have significantly improved our manuscript. We have made responding changes to the Discussion section. This change can be found – page 12, line 279-289.]

Comments 17: [258-259 – delete this sentence. It has nothing to do with your results.]

Response 17: [Thank for your professional suggestions, which have significantly improved our manuscript. We have made responding changes to the Discussion section.]

Comments 18: [260-269 – This part is more suitable for the Introduction.]

Response 18: [Thank for your professional suggestions, which have significantly improved our manuscript. We have made responding changes to the Introduction section. This change can be found – page 2, line 82-89.]

Comments 19: [270 – high-nutrient wetlands. What nutrient content is high (or less)?]

Response 19: [Sorry for the confusion. First of all, thank you very much for taking the valuable time to review. The total nitrogen, total phosphorus, and Chlorophyll-a concentrations of high-nutrient wetlands were high.]

Comments 20: [275 – Brachionus – italics.]

Response 20: [Thank for your professional suggestions, which have significantly improved our manuscript. We have made responding changes to the Results section. This change can be found – page 12, line 295.]

Comments 21: [287 – “Each zooplankton functional group responded differently to environmental factors”. Functional groups form specific species. Therefore, it is necessary to indicate specific species, and not just groups.]

Response 21: [Thank for your professional suggestions, which have significantly improved our manuscript. We have made responding changes to the Results section. This change can be found – page 11, line 309, 315, 321.]

Comments 22: [Tables S1 and S2 are small and can be placed in the manuscript. I recommend merging these tables. There are no needs for columns with Feeding habit (S1) and Taxonomic group (S2). It is possible to distribute zooplankton species into functional groups. In addition, there are a number of inaccuracies in the names of species. Many names are not valid and unaccepted. Some inaccuracies are marked in color directly in the table. For the future, it should be taken into account that now the taxonomy is changing significantly, so the species composition needs to be checked (for example, such sites as WoRMS, GBIF etc.)]

Response 22: [Thank for your professional suggestions, which have significantly improved our manuscript. We have made responding changes to the Materials and Methods section 2.2. This change can be found – page 7-9, line 234. 

The Polyarthra trigla and Limnoithona sinensis can be found in GBIF. (https://www.gbif.org/searchq=Polyarthra%20trigla,https://www.gbif.org/search?q=Limnoithona%20sinensis).]

Reviewer 2 Report

Comments and Suggestions for Authors

Dear Authors.

Main question.

Line 189 and Table 1. SD in wetlands varied 60.58 (± 4.81) m. This is mistake or that? The highest transparency in freshwater bodies is in Lake Baikal about 40 m. You should explain and correct these data.

Reviewer 3 Report

Comments and Suggestions for Authors

Review for the paper "How do zooplankton communities respond to environmental factors across the subsidence wetlands created by underground coal mining in the North China Plain?" by Yue Liang, Jianjun Huo, Weiqiang Li, Yutao Wang, Guangyao Wang, Chunlin Li submitted to "Diversity".

General comment.

Zooplankton play a critical ecological role in aquatic environments, providing a direct link between primary producers and higher trophic levels. Although it is generally accepted that coupling between physical and biological processes modulates the abundance and species composition of zooplankton in freshwater and wetlands, the dominant scales of these interactions are poorly understood. Most of the available information on plankton-environment interactions comes from studies in temperate waters, which are characterized by strong winter/summer seasonality in heat input. The increasing anthropogenic impact on ecosystems has become one of the most challenging problems in recent decades. Industrial human activities can create new habitats for various biotic communities. One of the examples are artificial water bodies, which serve as a new ecosystem for plankton assemblages. The authors aimed to investigate zooplankton-environment relationships in wetlands created as a result of coal mining in China. They found different responses of common zooplankton groups to the ecological factors. The study provides interesting results that can be helpful in aquatic monitoring and assessment of wetland ecosystem health. However, there are some issues that need to be addressed before the paper can be accepted.

Major flaws.

Section 3.2. Community structure is not well studied. It is suggested to perform clustering of NMDS to show spatial pattern in zooplankton. Furthermore, dominant and indicator species need to be defined and their contribution to the total zooplankton abundance needs to be presented in the results.

Section 3.2. It is suggested to compare densities of different functional groups with ANOVA or Kruskal-Wallis test.

Discussion. The environmental data should be briefly discussed. Were they typical? what about climatic changes in the region?

Discussion. Composition and community structure need to be discussed in more detail.

Conclusion. Projected changes in artificial wetlands must be proposed, and authors must clearly emphasize the ecological implications of their results for other wetland ecosystems.

Specific remarks.

L18 and Section 2.4. The statements are contradictory. In the Abstract, the authors mention generalized linear mixed models, while in the M&M and in the Results they refer to GLM. Please correct.

L 27. Replace "biological taxa" with "taxa" throughout the entire text.

Coordinates must be added to Figure 1.

Section 2.2. L 131. Specify the opening area of the net.

L136-137. The procedure for estimating zooplankton abundance must be described in detail.

L140. Consider replacing "crustaceans carnivora" with crustaceans carnivores".

L185-186. The pH is usually displayed without a unit. It is recommended to delete mg/L for pH values

L187 and Table 1. I am not comfortable that the authors have provided the appropriate unit for Chl-a. It should be mg m-3 or μg L-1.

L189-195. The unit for square should be corrected. "2" must be in the upper index.

Comments on the Quality of English Language

Minor revision

Author Response

Comments 1: [Section 3.2. Community structure is not well studied. It is suggested to perform clustering of NMDS to show spatial pattern in zooplankton. Furthermore, dominant and indicator species need to be defined and their contribution to the total zooplankton abundance needs to be presented in the results.]

Response 1: [Sorry for the confusion. First of all, thank you very much for taking the valuable time to review. We have made responding changes to the Methods section 2.4 and Result section 3.2. This change can be found – page 5, line 185-192; page 6, line 214-225. In this research, we randomly selected 34 subsidence wetlands on the North China Plain, which are spatially at the same latitude, so we did not group the 34 wetlands more deeply, so we did not use NMDS and indicator species analysis for detailed analysis. We acknowledge that the NMDS and indicator species analysis are good methods for analyzing the spatial pattern of zooplankton, and We will collect the data in other latitudes and classify in the near future and analyze the seasonal differences spatial pattern of zooplankton.

Thanks again for your careful review and constructive suggestions. We have made corresponding modifications according to your comments and those of the other two reviewers, and we hope that our revised manuscript may be approved by you.] 

Comments 2: [Section 3.2. It is suggested to compare densities of different functional groups with ANOVA or Kruskal-Wallis test.]

Response 2: [Thank for your professional suggestions, which have significantly improved our manuscript. We have made responding changes to the Methods section. This change can be found – page 5, line 185-187; page 6, line 214-215. The significant differences between the densities of the different functional groups of zooplankton (Kruskal-Wallis chi-squared = 209.69, df = 6, p-value < 2.2e-16).]

Comments 3: [Discussion. The environmental data should be briefly discussed. Were they typical? what about climatic changes in the region?]

Response 3: [Thank for your professional suggestions, which have significantly improved our manuscript. We have made responding changes to the Discussion section. We found all the subsidence wetlands were weakly alkaline, similar to other wetlands in the same region [1,2]. The wetlands had high concentrations of nitrogen and phosphorus, similar to other subsidence wetlands in the North China Plain [3]. The regional climate is warm temperate monsoon climate and the annual rainfall reaches 970 mm. This change can be found – page 11, line 279-284.

1. Chi, S.; Hu, J.; Li, M.; Wang, C. What Are the Relationships between Plankton and Macroinvertebrates in Reservoir Systems? Water. 2023, 15, 2682.

2. Fan, T.; Amzil, H.; Fang, W.; Xu, L.; Lu, A.; Wang, S.; Wang, X.; Chen, Y.; Pan, J.; Wei, X. Phytoplankton-Zooplankton Community Structure in Coal Mining Subsidence Lake. Int J Env Res Pub He. 2022, 20, 484.

3. Yi, Q.; Wang, X.; Wang. T.; Qu, X.; Xie, K. Eutrophication and nutrient limitation in the aquatic zones around Huainan coal mine subsidence areas, Anhui, China. Water Sci Technol. 2014, 70, 878-887.

Thanks again for your careful review and constructive suggestions. We have made corresponding modifications according to your comments and those of the other two reviewers, and we hope that our revised manuscript may be approved by you.]

Comments 4: [Discussion. Composition and community structure need to be discussed in more detail.]

Response 4: [Thank for your professional suggestions, which have significantly improved our manuscript. We have made responding changes to the Discussion section. This change can be found – page 12, line 286-289.]

Comments 5: [Conclusion. Projected changes in artificial wetlands must be proposed, and authors must clearly emphasize the ecological implications of their results for other wetland ecosystems.]

Response 5: [Thank for your professional suggestions, which have significantly improved our manuscript. We have made responding changes to the Conclusion section. This change can be found – page 13, line 359-361.]

Comments 6: [L18 and Section 2.4. The statements are contradictory. In the Abstract, the authors mention generalized linear mixed models, while in the M&M and in the Results they refer to GLM. Please correct.]

Response 6: [Thank for your professional suggestions, which have significantly improved our manuscript. We have made responding changes to the Abstract section. This change can be found – page 1, line 18.]

Comments 7: [L 27. Replace "biological taxa" with "taxa" throughout the entire text.]

Response 7: [Thank for your professional suggestions, which have significantly improved our manuscript. We have made responding changes to the introduction section. This change can be found – page 1, line 36, 37, 40, 43; page 2, line 63, 64, 72.]

Comments 8: [Coordinates must be added to Figure 1.]

Response 8: [Thank for your professional suggestions, which have significantly improved our manuscript. We have made responding changes to the Methods section 2.1. This change can be found – page 3, line 117-118.]

Comments 9: [Section 2.2. L 131. Specify the opening area of the net.]

Response 9: [Thank for your professional suggestions, which have significantly improved our manuscript. We have made responding changes to the Methods section. This change can be found – page 4, line 134.]

Comments 10: [L136-137. The procedure for estimating zooplankton abundance must be described in detail.]

Response 10: [Thank for your professional suggestions, which have significantly improved our manuscript. We have made responding changes to the Methods section 2.2. This change can be found – page 4, line 139-141.]

Comments 11: [L140. Consider replacing "crustaceans carnivora" with crustaceans carnivores".]

Response 11: [Thank for your professional suggestions, which have significantly improved our manuscript. We have made responding changes to the Methods section. This change can be found – page 4, line 145.]

Comments 12: [L185-186. The pH is usually displayed without a unit. It is recommended to delete mg/L for pH values]

Response 12: [Thank for your professional suggestions, which have significantly improved our manuscript. We have made responding changes to the Methods section. This change can be found – page 5, line 197.]

Comments 13: [L187 and Table 1. I am not comfortable that the authors have provided the appropriate unit for Chl-a. It should be mg m-3 or μg L-1.]

Response 13: [Thank for your professional suggestions, which have significantly improved our manuscript. We have made responding changes to the Result section 3.1, Table 2. This change can be found – page 7, line 195.]

Comments 14: [L189-195. The unit for square should be corrected. "2" must be in the upper index.]

Response 14: [Thank for your professional suggestions, which have significantly improved our manuscript. We have made responding changes to the Methods section 3.1. This change can be found – page 7, line 196-200.]

Round 2

Reviewer 2 Report

Comments and Suggestions for Authors

Line 23. It is possible, to stress the impact of the photovoltaic panel area­ " and mostly negatively correlated with the photovoltaic panel area"

Lime 169 and further in the text Y = (Ni/N). i it should be in lower case.

Reviewer 3 Report

Comments and Suggestions for Authors

The authors have revised the paper according to my comments.